# SMiRL: Surprise Minimizing RL in Dynamic Environments

## Abstract

All living organisms struggle against the forces of nature to carve out niches where they can maintain relative stasis. We propose that such a search for order amidst chaos might offer a unifying principle for the emergence of useful behaviors in artificial agents. We formalize this idea into an unsupervised reinforcement learning method called surprise minimizing RL (SMiRL). SMiRL trains an agent with the objective of maximizing the probability of observed states under a model trained on all previously seen states. The resulting agents acquire several proactive behaviors to seek and maintain stable states such as balancing and damage avoidance, that are closely tied to the environment's prevailing sources of entropy, such as winds, earthquakes, and other agents. We demonstrate that our surprise minimizing agents can successfully play Tetris, Doom, and control a humanoid to avoid falls, without any task-specific reward supervision. We further show that SMiRL can be used together with a standard task reward to accelerate reward-driven learning.

## 1 Introduction

> *The general struggle for existence of animate beings is not a struggle for raw materials, nor for energy, but a struggle for negative entropy.*
>
> (Ludwig Boltzmann, 1886)

All living organisms carve out environmental niches within which they can maintain relative predictability amidst the ever-increasing entropy around them (Boltzmann, 1886; Schrödinger, 1944; Schneider & Kay, 1994; Friston, 2009). Humans, for example, go to great lengths to shield themselves from surprise — we band together in millions to build cities with homes, supplying water, food, gas, and electricity to control the deterioration of our bodies and living spaces amidst heat and cold, wind and storm. The need to discover and maintain such surprise-free equilibria has driven great resourcefulness and skill in organisms across very diverse natural habitats. Motivated by this, we ask: could the motive of preserving order amidst chaos guide the automatic acquisition of useful behaviors in artificial agents?

Our method therefore addresses the unsupervised reinforcement learning problem: how might an agent in an environment acquire complex behaviors and skills with no external supervision? This central problem in artificial intelligence has evoked several candidate solutions, largely focusing on novelty-seeking behaviors (Schmidhuber, 1991; Lehman & Stanley, 2011; Still & Precup, 2012; Bellemare et al., 2016; Houthooft et al., 2016; Pathak et al., 2017). In simulated worlds, such as video games, novelty-seeking intrinsic motivation can lead to interesting and meaningful behavior. However, we argue that these sterile environments are fundamentally lacking compared to the real world. In the real world, natural forces and other agents offer bountiful novelty. The second law of thermodynamics stipulates ever-increasing entropy, and therefore perpetual novelty, without even requiring any agent intervention. Instead, the challenge in natural environments is homeostasis: discovering behaviors that enable agents to maintain an equilibrium, for example to preserve their bodies, their homes, and avoid predators and hunger. Even novelty seeking behaviors may emerge naturally as a means to maintain homeostasis: an agent that is curious and forages for food in unlikely places might better satisfy its hunger.

Figure 1: In natural environments (left), an inactive agent will experience a wide variety of states. By reasoning about future surprise, a SMiRL agent can take actions that temporarily increase surprise but reduce it in the long term. For example, building a house initially results in novel states, but once it is built, the house allows the agent to experience a more stable and surprise-free environment. On the right we show an interpretation of the agent interaction loop using SMiRL. When the agent observes a state, it updates it belief $p(\mathbf{s})$ over states. Then, the action policy $\pi(a|\mathbf{s}, \theta)$ is conditioned on this belief and maximizes the expected likelihood of the next state under its belief.

We formalize allostasis as an objective for reinforcement learning based on surprise minimization (SMiRL). In highly entropic and dynamic environments with undesirable forms of novelty, minimizing surprise (i.e., minimizing novelty) causes agents to naturally seek a stable equilibrium. Natural environments with winds, earthquakes, adversaries, and other disruptions already offer a steady stream of novel stimuli, and an agent that minimizes surprise in these environments will act and explore in order to find the means to maintain a stable equilibrium in the face of these disturbances.

SMiRL is simple to describe and implement: it works by maintaining a density $p(\mathbf{s})$ of visited states and training a policy to act such that future states have high likelihood under $p(\mathbf{s})$. This interaction scheme is shown in Figure 1(right) Across many different environments, with varied disruptive forces, and in agents with diverse embodiments and action spaces, we show that this simple approach induces useful equilibrium-seeking behaviors. We show that SMiRL agents can solve Tetris, avoid fireballs in Doom, and enable a simulated humanoid to balance and locomote, without any explicit task reward. More pragmatically, we show that SMiRL can be used together with a task reward to accelerate standard reinforcement learning in dynamic environments, and can provide a simple mechanism for imitation learning. SMiRL holds promise for a new kind of unsupervised RL method that produces behaviors that are closely tied to the prevailing disruptive forces, adversaries, and other sources of entropy in the environment. Videos of our results are available at `https://sites.google.com/view/surpriseminimization`

## 2 SURPRISE MINIMIZING AGENTS

We propose surprise minimization as a means to operationalize the idea of learning useful behaviors by seeking to preserve order amidst chaos. In complex natural environments with disruptive forces that tend to naturally increase entropy, which we refer to as *entropic* environments, minimizing surprise over an agent's lifetime requires taking action to reach stable states, and often requires acting continually to maintain homeostasis and avoid surprise. The long term effects of actions on the agent's surprise can be complex and somewhat counterintuitive, especially when we consider that actions not only change the state that the agent is in, but also *its beliefs* about which states are more likely. The combination of these two processes induce the agent to not only seek states where $p(\mathbf{s})$ is large, but to also visit states so as to alter $p(\mathbf{s})$, in order to receive larger rewards in the future. This "meta" level reasoning can result in behaviors where the agent might actually visit new states *in order to make them more familiar*. An example of this is shown in Figure 1 where in order to avoid the disruptions from the changing weather an agent needs to build a shelter or home to protect itself and decrease its observable surprise. The SMiRL formulation relies on disruptive forces in the environment to avoid collapse to degenerate solutions, such as staying in a single state $\mathbf{s}_0$. Fortunately, natural environments typically offer no shortage of such disruption.

### 2.1 SURPRISE MINIMIZATION PROBLEM STATEMENT

To instantiate SMiRL, we design a reinforcement learning agent with a reward proportional to how familiar its current state is based on the history of states it has experienced during its "life," which corresponds to a single episode. Formally, we assume a fully-observed controlled Markov process

(CMP), though extensions to partially observed settings can also be developed. We use $\mathbf{s}_t$ to denote the state at time $t$, and $a_t$ to denote the agent's action, $\rho(s_0)$ to denote the initial state distribution, and $T(\mathbf{s}_{t+1}|\mathbf{s}_t, a_t)$ to denote the transition dynamics. The agent has access to a dataset $\mathcal{D}_t = \{\mathbf{s}_1, \ldots, \mathbf{s}_t\}$ of all states experienced so far. By fitting a generative model $p_{\theta_t}(\mathbf{s})$ with parameters $\theta_t$ to this dataset, the agent obtains an estimator that can be used to evaluate the negative surprise reward, given by

$$r_t(\mathbf{s}) = \log p_{\theta_t}(\mathbf{s}) \tag{1}$$

We denote the fitting process as $\theta_t = \mathcal{U}(\mathcal{D}_t)$. The goal of a SMiRL agent is to maximize the sum $\sum_t \log p_{\theta_t}(\mathbf{s}_{t+1})$. Since the agent's actions affect the future $\mathcal{D}_t$ and thus the future $\theta_t$'s, the optimal policy does not simply visit states that have a high $p_{\theta_t}(\mathbf{s})$ now, but rather those states that will change $p_{\theta_t}(\mathbf{s})$ such that it provides high likelihood to the states that it sees in the future.

## 2.2 TRAINING SMiRL AGENTS

We now present a practical reinforcement learning algorithm for surprise minimization. Recall that a critical component of SMiRL is reasoning about the effect of actions on *future* states that will be added to $\mathcal{D}$, and their effect on *future* density estimates – e.g., to understand that visiting a state that is currently unfamiliar and staying there will make that state familiar, and therefore lead to higher rewards in the long run. This means that the agent must reason not only about the unknown MDP dynamics, but also the dynamics of the density model $p_\theta(\mathbf{s})$ trained on $\mathcal{D}$. In our algorithm, we accomplish this via an episodic training

---

**Algorithm 1** Training a SMiRL agent with RL

1: Initialize policy parameters $\phi$
2: Initialize RL algorithm RL
3: **for each** episode $= 1, 2, \ldots$ **do**
4:     $\mathbf{s}_0 \sim \rho(\mathbf{s}_0)$       ▷ Initial state distribution.
5:     $\mathcal{D}_0 \leftarrow \{s_0\}$       ▷ Reset state history.
6:     **for each** $t = 0, 1, \ldots, T$ **do**
7:         $\theta_t \leftarrow \mathcal{U}(\mathcal{D}_t)$     ▷ Fit density model.
8:         $a_t \sim \pi_\phi(a_t|\mathbf{s}_t, \theta_t, t)$   ▷ Run policy.
9:         $\mathbf{s}_{t+1} \sim T(\mathbf{s}_{t+1}|\mathbf{s}_t, a_t)$ ▷ Transition dynamics.
10:       $r_t \leftarrow \log p_{\theta_t}(\mathbf{s}_{t+1})$   ▷ Familiarity reward.
11:       $\mathcal{D}_{t+1} \leftarrow \mathcal{D}_t \cup \{\mathbf{s}_{t+1}\}$  ▷ Update state history.
12:     **end for each**
13:     $\phi \leftarrow \text{RL}(\phi, \mathbf{s}_{[0:T]}, \theta_{[0:T]}, |\mathcal{D}|_{[0:T]}, a_{[0:T]}, r_{[0:T]})$
14: **end for each**

---

procedure, where the agent is trained over many episodes and $\mathcal{D}$ is reset at the beginning of each episode to simulate a new lifetime. Through this procedure, SMiRL learns the parameters $\phi$ of the agent's policy $\pi_\phi$ for a fixed horizon. To learn this the policy must be conditioned on some sufficient statistic of $\mathcal{D}_t$, since the reward $r_t$ is a function of $\mathcal{D}_t$.

Having trained parameterized generative models $p_{\theta_t}$ as above on all states seen so far, we condition $\pi$ on $\theta_t$ and $|\mathcal{D}_t|$. This implies an assumption that $\theta_t$ and $|\mathcal{D}_t|$ represent the sufficient statistics necessary to summarize the contents of the dataset for the policy, and contain all information required to reason about how $p_\theta$ will evolve in the future. Of course, we could also use any other summary statistic, or even read in the entirety of $\mathcal{D}_t$ using a recurrent model. In the next section, we also describe a modification that allows us to utilize a deep density model without conditioning $\pi$ on a high-dimensional parameter vector.

Algorithm 1 provides the pseudocode. SMiRL can be used with any reinforcement learning algorithm, which we denote RL in the pseudocode. As is standard in reinforcement learning, we alternate between sampling episodes from the policy (lines 6-12) and updating the policy parameters (line 13). The details of the updates are left to the specific RL algorithm, which may be on or off-policy. During each episode, as shown in line 11, $\mathcal{D}_0$ is initialized with the first state and grows as each state visited by the agent is added to the dataset. The parameters $\theta_t$ of the density model are fit to $\mathcal{D}_t$ at each timestep to both be passed to the policy and define the reward function. At the end of the episode, $\mathcal{D}_T$ is discarded and the new $\mathcal{D}_0$ is initialized.

## 2.3 STATE DENSITY ESTIMATION WITH LEARNED REPRESENTATIONS

While SMiRL may in principle be used with any choice of model class for the generative model $p_\theta(\mathbf{s})$, this choice must be carefully made in practice. As we show in our experiments, relatively simple distribution classes, such as products of independent marginals, suffice to run SMiRL in simple environments with low-dimensional state spaces. However, it may be desirable in more complex

environments to use more sophisticated density estimators, especially when learning directly from high-dimensional observations such as images.

In particular, we propose to use variational autoencoders (VAEs) (Kingma & Welling, 2014) to learn a non-linear compressed state representation and facilitate estimation of $p_\theta(\mathbf{s})$ for SMiRL. A VAE is trained using the standard loss to reconstruct states $\mathbf{s}$ after encoding them into a low-dimensional normal distribution $q_\omega(\mathbf{z}|\mathbf{s})$ through the encoder $q$ with parameters $\omega$. A decoder $p_\psi(\mathbf{s}|\mathbf{z},)$ with parameters $\psi$ computes $\mathbf{s}$ from the encoder output $\mathbf{z}$. During this training process, a KL divergence loss between the prior $p(\mathbf{z})$ and $q_\omega(\mathbf{z}|\mathbf{s})$ is used to keep this distribution near the standard normal distribution. We described a VAE-based approach for estimating the SMiRL surprise reward. In our implementation, the VAE is trained online, with VAE updates interleaved with RL updates. Training a VAE requires more data than the simpler density models that can easily be fit to data from individual episodes. We propose to overcome this by not resetting the VAE parameters between training episodes. Instead, we train the VAE across episodes. Instead of passing all VAE parameters to the SMiRL policy, we track a separate episode-specific distribution $p_{\theta_t}(\mathbf{z})$, distinct from the VAE prior, over the course of each episode. $p_{\theta_t}(\mathbf{z})$ replaces $p_{\theta_t}(\mathbf{s})$ in the SMiRL algorithm and is fit to only that episode's state history. We represent $p_{\theta_t}(\mathbf{z})$ as a vector of independent normal distributions , and fit it to the VAE encoder outputs. This replaces the density estimate in line 10 of Algorithm 1. Specifically, the corresponding update $\mathcal{U}(\mathcal{D}_t)$ is performed as follows:

$$\mathbf{z}_0, \ldots, \mathbf{z}_t = \mathbb{E}[q_\omega(\mathbf{z}|\mathbf{s})] \text{ for } \mathbf{s} \in \mathcal{D}_t$$
$$\mu = \frac{\sum_{j=0}^t \mathbf{z}_j}{t+1}, \sigma = \frac{\sum_{j=0}^t (\mu - \mathbf{z}_j)^2}{t+1}$$
$$\theta_t = [\mu, \sigma].$$

Training the VAE online, over all previously seen data, deviates from the recipe in the previous section, where the density model was only updated *within* an episode. However, this does provide for a much richer state density model, and the within-episode updates to estimate $p_{\theta_t}(\mathbf{z})$ still provide our method with meaningful surprise-seeking behavior. As we show in our experiments, this can improve the performance of SMiRL in practice.

## 3 ENVIRONMENTS

We evaluate SMiRL on a range of environments, from video game domains to simulated robotic control scenarios. These are rich, *dynamic* environments — the world evolves automatically even without agent intervention due to the presence of disruptive forces and adversaries. Note that SMiRL relies on such disruptions to produce meaningful emergent behavior, since mere inaction would otherwise suffice to achieve homeostasis. However, as we have argued above, such disruptions are also an important property of most real world environments. Current RL benchmarks neglect this, focusing largely on unrealistically sterile environments where the agent alone drives change (Bellemare et al., 2015; Brockman et al., 2016). Therefore, our choices of environments, discussed below, are not solely motivated by suitability to SMiRL; rather, we aim to evaluate unsupervised RL approaches, ours as well as others, in these more dynamic environments.

***Tetris.*** The classic game of *Tetris* offers a naturally entropic environment — the world evolves according to its own rules and dynamics even in the absence of coordinated behavior of the agent, piling up pieces and filling up the board. It therefore requires active intervention to maintain homeostasis. We consider a $4 \times 10$ *Tetris* board with tromino shapes (composed of 3 squares), as shown in Figure 2a. The observation is a binary image of the current board with one pixel per square, as well as an indicator for the type of shape that will appear next. Each action denotes one of the 4 columns in which to drop the shape and one of 4 shape orientations. For evaluation, we measure how many rows the agent clears, as well as how many times the agent dies in the game by allowing the blocks to reach the top of the board, within the max episode length of 100. Since the observation is a binary image, we model $p(\mathbf{s})$ as independent Bernoulli. See Appendix A for details.

***VizDoom.*** We consider two *VizDoom* environments from Kempka et al. (2016): *TakeCover* and *DefendTheLine*. *TakeCover* provides a dynamically evolving world, with enemies that appear over time and throw fireballs aimed at the player (Kempka et al., 2016). The observation space consists of the 4 previous grayscale first-person image observations, and the action space consists of moving left or right. We evaluate the agent based on how many times it is hit by fireballs, which we term

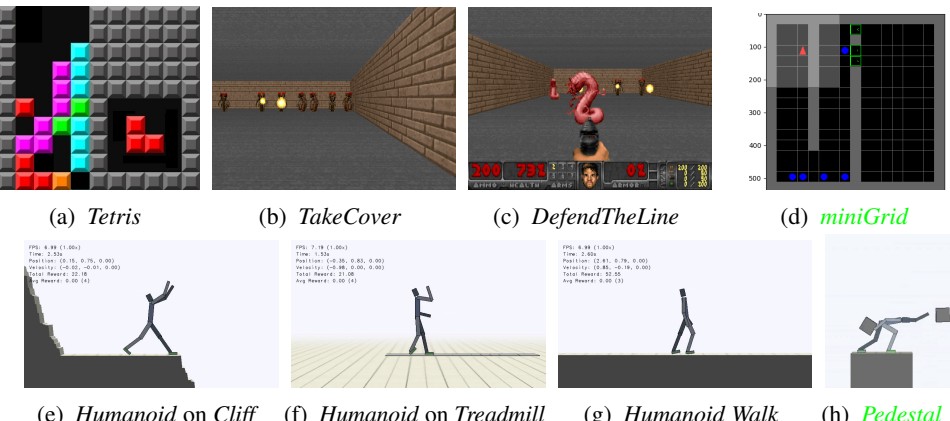

| (a) *Tetris* | (b) *TakeCover* | (c) *DefendTheLine* | (d) *miniGrid* |

| (e) *Humanoid* on *Cliff* | (f) *Humanoid* on *Treadmill* | (g) *Humanoid Walk* | (h) *Pedestal* |

Figure 2: Illustrations of the entropic evaluation environments: (a) A rendering of the *Tetris* environment. (b) The *TakeCover* environment, with enemies throwing fireballs in the distance. (c) *VizDoom DefendTheLine* environment with multiple enemies. (d) is the *miniGrid* environment with one agent (in red) and a number of enemeies (blue). (e) The simulated *Humanoid* next to a cliff. (f) The *Humanoid* on a treadmill, (g) a *Humanoid* learning to walk and (h) *Pedestal*.

the "damage" taken by the agent. Images from the *TakeCover* environment are shown in Fig 2b and Fig **??**.

In *DefendTheLine*, additional enemies can move towards the player, and the player can shoot the enemies. The agent starts with limited ammunition. This environment provides a "survival" reward function ($r = 1$ for each timestep alive) and performance is measured by how long the agent survives in the environment. For both environments, we model $p(\mathbf{s})$ as independent Gaussian over the pixels. See Appendix A for details.

*miniGrid* Is a navigation task where the agent has a partial observation of the environment shown by the lighter gray area around the red agent in Figure 2d. The agent needs to navigate down the hallways to escape the enemy agents (blue) to reach the *safe* room on the right the enemies can not enter, through a randomly placed door.

**Simulated *Humanoid* robots.** In the last set of environments, a simulated planar *Humanoid* robot is placed in situations where it is in danger of falling. The action consists of the PD targets for each of the joints. The state space comprises the rotation of each joint and the linear velocity of each link. We evaluate several versions of this task, which are shown in Figure 2. The *Cliff* tasks starts the agent at the edge of a cliff, in a random pose and with a forward velocity of 1 m/s. Falling off the cliff leads to highly irregular and unpredictable configurations, so a surprise minimizing agent will want to learn to stay on the cliff. In the *Treadmill* environment, the robot starts on a platform that is moving at 1 m/s backwards; an agent will be carried backwards unless it learns some locomotion. The *Pedestal* environment is designed to show that SMiRL can learn a more active balancing policy. In this environment the agent starts out on a thin pedestal and random forces are applied to the robots links and boxes of random size are thrown at the agent. The *Walk* domain is used to evaluate the use of the SMiRL reward as a form of "stability reward" that assists the agent in learning how to walk while reducing the number of falls. This is done by initializing $p(\mathbf{s})$ from example walking data and adding this to the task reward, as discussed in Section 4.2. The task reward in *Walk* is $r_{\mathtt{walk}} = exp((v_d * v_d) * -1.5)$, where $v_d$ is the difference between the $x$ velocity and the desired velocity of 1 m/s. In these environments, we measure performance as the proportion of episodes with a fall. A state is classified as a fall if either the agent's links, except for the feet, are touching the ground, or if the agent is $-5$ meters or more below the level of the platform or cliff. Since the state is continuous, we model $p(\mathbf{s})$ as independent Gaussian; see Appendix A for details.

## 4 EXPERIMENTAL RESULTS

Our experiments aim to answer the following questions: **(1)** Can SMiRL learn meaningful and complex emergent behaviors in the environments described in Section 3? **(2)** Can we incorporate

deep generative models into SMiRL and use state densities in learned representation spaces? **(3)** Can SMiRL serve as a joint training objective to accelerate acquisition of reward-guided behavior, and does it outperform prior intrinsic motivation methods in this role? We also illustrate several applications of SMiRL, showing that it can accelerate task learning, provide for exploration with fewer damaging falls, and provide for elementary imitation. Videos of learned behaviors are available on the website `https://sites.google.com/view/surpriseminimization/home`

### 4.1 EMERGENT BEHAVIOR IN UNSUPERVISED LEARNING

First, we evaluate SMiRL on the *Tetris*, *VizDoom*, *Cliff*, and *Treadmill* tasks, studying its ability to generate purposeful coordinated behaviors after training using only the surprise minimizing objective, in order to answer question **(1)**. The SMiRL agent demonstrates meaningful emergent behaviors in each of these domains. In the *Tetris* environment, the agent is able to learn proactive behaviors to eliminate rows and properly play the game. The agent also learns emergent game playing behaviour in the *VizDoom* environment, acquiring an effective policy for dodging the fireballs thrown by the enemies. In both of these environments, stochastic and chaotic events force the SMiRL agent to take a coordinated course of action to avoid unusual states, such as full Tetris boards or fireball explosions. In the *Cliff* environment, the agent learns a policy that greatly reduces the probability of falling off of the cliff by bracing against the ground and stabilize itself at the edge, as shown in Figure 2e. In the *Treadmill* environment, SMiRL learns a more complex locomotion behavior, jumping forward to increase the time it stays on the treadmill, as shown in Figure 2f. A quantitative measurement of the reduction in falls is shown in Figure 4.

We also study question **(2)** in the *TakeCover*, *Cliff*, *Treadmill* and *Pedestal* environments, training a VAE model and estimating surprise in the latent space of the VAE. In most of these environments, the representation learned by the VAE leads to faster acquisition of the emergent behaviors in *Take-Cover* Figure 3 (right), *Cliff* Figure 4 (left), and *Treadmill* Figure 4 (middle), leads to a substantially more successful locomotion behavior.

**Comparison to intrinsic motivation.** Figure 3 shows plots of the environment-specific rewards over time on *Tetris*, *TakeCover*, and the *Humanoid* domains Figure 4. In order to compare SMiRL to more standard intrinsic motivation methods, which seek out states that *maximize* surprise or novelty, we also evaluated ICM (Pathak et al., 2017) and RND (Burda et al., 2018b). We also plot an oracle agent that directly optimizes the task reward. On *Tetris*, after training for 2000 epochs, SMiRL achieves near perfect play, on par with the oracle reward optimizing agent, with no deaths, as shown in Figure 3 (left, middle). ICM seeks novelty by creating more and more distinct patterns of blocks rather than clearing them, leading to deteriorating game scores over time. On *TakeCover*, SMiRL effectively learns to dodge fireballs thrown by the adversaries, as shown in 3 (right). Novelty-seeking ICM once again yields deteriorating rewards over time due to the method seeking novel events that correspond to damage. The baseline comparisons for the *Cliff* and *Treadmill* environments have a similar outcome. The novelty seeking behaviour of ICM causes it to learn a type of irregular behaviour that causes the agent to jump off the *Cliff* and roll around on the *Treadmill*, maximizing the variety (and quantity) of falls Figure 4.

SMiRL and curiosity are not mutually exclusive. We show that these intrinsic reward functions can be combined to achieve better results on the *Treadmill* environment Figure 4(right). The combination of methods leads to increased initial learning speed and producing a walking-type gait on that task.

**Exploration for SMiRL** To illustrate SMiRL's desire to explore we evaluate over an environment where the agent needs to produce long term planning behaviour. This environment is shown in Figure 2d, where the agent needs to navigate its way through the hallways, avoiding enemies, to reach a *safe* room through a randomly placed door. We found that SMiRL is able to solve this task. Results from these examples are shown on the accompanying website.

### 4.2 APPLICATIONS OF SMIRL

While the central focus of this paper is the emergent behaviors that can be obtained via SMiRL, in this section we study more pragmatic applications. We show that SMiRL can be used for joint training to accelerate reward-driven learning of tasks, and also illustrate how SMiRL can be used to produce a rudimentary form of imitation learning.

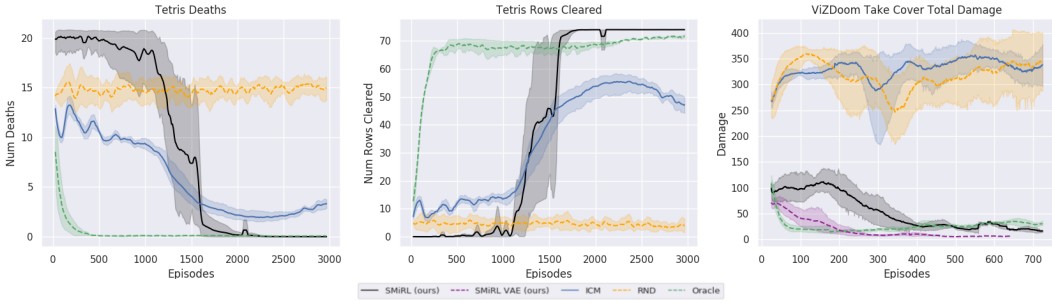

Figure 3: Results for video game environments: Comparison between SMiRL, ICM, RND, and an oracle RL algorithm with access to the true reward in *Tetris* on (left) number of deaths per episode (lower is better), (center) number of rows cleared per episode (higher is better), and (right) in *TakeCover* on amount of damage taken (lower is better). The SMiRL agent is able to learn how to play *Tetris* and avoid fireballs in *TakeCover* almost as well as an agent trained on the task reward. Using VAE features for the density model (SMiRL VAE) improves performance in *VizDoom*. Five random seeds are sampled for each method on each plot, the mean and standard deviation are shown. Videos of the policies can be found at: https://sites.google.com/view/surpriseminimization

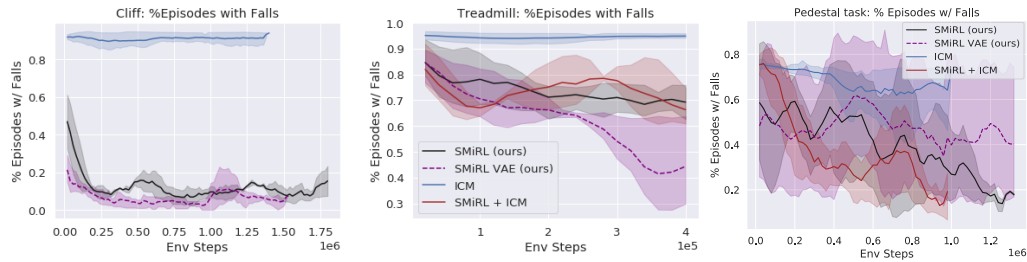

Figure 4: Results for the *Cliff*, *Treadmill* and *Pedestal* environments. In all cases, the SMiRL reward function reduces the fraction of episodes that results in falls (lower is better). The use of a VAE to estimate $p(\mathbf{s})$ often increases learning speed and final performance. Three random seeds are sampled for each method on each plot, the mean and standard deviation are shown.

**Imitation.** We can easily adapt SMiRL to perform imitation by initializing the buffer $\mathcal{D}_0$ with states from expert demonstrations, or even individual desired outcome states. To study this application of SMiRL, we initialize the buffer $\mathcal{D}_0$ in *Tetris* with user-specified *desired* board states. An illustration of the *Tetris* imitation task is presented in Figure 6, showing imitation of a box pattern (top) and a checkerboard pattern (bottom), with the leftmost frame showing the user-specified example, and the other frames showing actual states reached by the SMiRL agent. While a number of prior works have studied imitation without example actions (Liu et al., 2018; Torabi et al., 2018a; Aytar et al., 2018; Torabi et al., 2018b; Edwards et al., 2018; Lee et al.), this capability emerges automatically in SMiRL, without any further modification to the algorithm.

**SMiRL as a stability reward.** In this next experiment, we study how SMiRL can accelerate acquisition of reward-driven behavior in environments that present a large number of possible actions leading to diverse but undesirable states. Such settings are common in real life: a car can crash in many different ways, a robot can drop a glass on the ground causing it to break in many ways, etc. While this is of course not the case for all tasks, many real-world tasks do require the agent to stabilize itself in a specific and relatively narrow set of conditions. Incorporating SMiRL into the learning objective in such settings can accelerate learning, and potentially improve safety during training, as the agent automatically learns to avoid anything that is unfamiliar. We study this application of SMiRL in the *DefendTheLine* task and the *Walk* task. In both cases, we use SMiRL to augment the task reward, such that the full reward is given by $r_{\text{combined}}(\mathbf{s}) = r_{\text{task}}(\mathbf{s}) + \alpha r_{\text{SMiRL}}(\mathbf{s})$, where $\alpha$ is chosen to put the two reward terms at a similar magnitude. In the *Walk* task, illustrated in Figure 2g, $p_\theta(\mathbf{s})$ is additionally initialized with 8 example walking trajectories (256 timesteps each), similarly to the imitation setting, to study how well SMiRL can incorporate prior knowledge into the stability reward (Reward + SMiRL (ours). We include another version that is not initialized with expert data (Reward + SMiRL (no-expert). We measure the number of falls during training, with and without

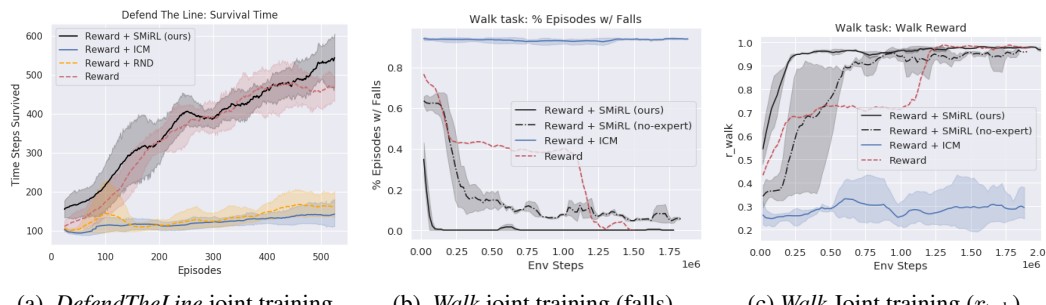

(a) *DefendTheLine* joint training    (b) *Walk* joint training (falls)    (c) *Walk* Joint training ($r_{\text{task}}$)

Figure 5: In (a) different intrinsic reward methods are combined with the survival time task reward in the *VizDoom DefendTheLine* task, showing that SMiRL accelerates learning compared to intrinsic motivation methods and the pure task reward. In (b) and (c) we combine the SMiRL reward with the *Walk* reward and initialize SMiRL with walking demonstrations and without (no-expert) . This results in significantly fewer falls (b) and faster learning w.r.t. the task reward (c). Five random seeds are sampled for (a) and three for (b and c), the mean and standard deviation are shown.

the SMiRL reward term. The results in Figure 5b show that adding the SMiRL reward results in significantly fewer falls during training, and less when using expert data while learning to walk well, indicating that SMiRL stabilizes the agent more quickly than the task reward alone.

In the *DefendTheLine* task, shown in Figure 2c, we compare the performance of SMiRL as a joint training objective to the more traditional novelty-driven bonus provided by ICM (Pathak et al., 2017) and RND (Burda et al., 2018b). Novelty-driven bonuses are often used to accelerate learning in domains that present an exploration challenge. However, as shown in the results in Figure 5a, the SMiRL reward, even without demonstration data, provides for substantially faster learning on this task than novelty-seeking intrinsic motivation. These results suggest that SMiRL can be a viable method for accelerating learning and reducing the amount of unsafe behavior (e.g., falling) in dynamic environments.

## 5    RELATED WORK

Prior works have sought to learn intelligent behaviors through reinforcement learning (Sutton & Barto, 2018) with respect to a provided reward function, such as the score of a video game (Mnih et al., 2013) or a hand-defined cost function (Levine et al., 2016). Such rewards are often scarce or difficult to provide in practical real world settings, motivating approaches for reward-free learning such as empowerment (Klyubin et al., 2005; Mohamed & Jimenez Rezende, 2015) or intrinsic motivation (Chentanez et al., 2005; Oudeyer & Kaplan, 2009; Oudeyer et al., 2007). Intrinsic motivation has typically focused on encouraging novelty-seeking behaviors by maximizing model uncertainty (Houthooft et al., 2016; Still & Precup, 2012; Shyam et al., 2018; Pathak et al., 2019), by maximizing model prediction error or improvement (Lopes et al., 2012; Pathak et al., 2017), through state visitation counts (Bellemare et al., 2016), via surprise maximization (Achiam & Sastry, 2017; Schmidhuber, 1991; Sun et al., 2011), and through other novelty-based reward bonuses (Lehman & Stanley, 2011; Burda et al., 2018a; Kim et al., 2019). We do the opposite. Inspired by the free energy principle (Friston, 2009; Friston et al., 2009), we instead incentivize an agent to *minimize* surprise and study the resulting behaviors in dynamic, entropy-increasing environments. In such environments, which we believe are more reflective of the real-world, we find that prior novelty-seeking environments perform poorly.

Prior works have also studied how competitive self-play and competitive, multi-agent environments can lead to complex behaviors with minimal reward information (Silver et al., 2017; Bansal et al., 2017; Sukhbaatar et al., 2017; Baker et al., 2019). Like these works, we also consider how complex behaviors can emerge in resource constrained environments. However, our approach can also be applied in non-competitive environments.

## 6 DISCUSSION

We presented an unsupervised reinforcement learning method based on *minimization* of surprise. We show that surprise minimization can be used to learn a variety of behaviors that maintain "homeostasis," putting the agent into stable and sustainable limit cycles in its environment. Across a range of tasks, these stable limit cycles correspond to useful, semantically meaningful, and complex behaviors: clearing rows in Tetris, avoiding fireballs in VizDoom, and learning to balance and hop forward with a bipedal robot. The key insight utilized by our method is that, in contrast to simple simulated domains, realistic environments exhibit dynamic phenomena that gradually increase entropy over time. An agent that resists this growth in entropy must take active and coordinated actions, thus learning increasingly complex behaviors. This stands in stark contrast to commonly proposed intrinsic exploration methods based on novelty, which instead seek to visit novel states and increase entropy.

Besides fully unsupervised reinforcement learning, where we show that our method can give rise to intelligent and complex policies, we also illustrate several more pragmatic applications of our approach. We show that surprise minimization can provide a general-purpose risk aversion reward that, when combined with task rewards, can improve learning in environments where avoiding catastrophic (and surprising) outcomes is desirable. We also show that SMiRL can be adapted to perform a rudimentary form of imitation.

Our investigation of surprise minimization suggests a number of directions for future work. The particular behavior of a surprise minimizing agent is strongly influenced by the particular choice of state representation: by including or excluding particular observation modalities, the agent will be more or less surprised. Thus, tasks may potentially be designed by choosing appropriate state or observation representations. Exploring this direction may lead to new ways of specifying behaviors for RL agents without explicit reward design. Other pragmatic applications of surprise minimization may also be explored in future work, including its effects for mitigating reward misspecification, by disincentivizing any unusual behavior that likely deviates from what the reward designer intended. Finally, we believe that a promising direction for future research is to study how lifelong surprise minimization can result in intelligent and sophisticated behavior that maintains homeostasis by acquiring increasingly complex behaviors. This may be particularly relevant in complex real-world environments populated by other intelligent agents, where maintaining homeostasis may require constant adaptation and exploration.

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

# A  IMPLEMENTATION DETAILS

**SMiRL on *Tetris*.**  In *Tetris*, since the state is a binary image, we model $p(\mathbf{s})$ as a product of independent Bernoulli distributions for each board location. The SMiRL reward $\log p_\theta(\mathbf{s})$ from (1) becomes:

$$r_{\text{SMiRL}}(\mathbf{s}) = \sum_i \mathbf{s}_i \log \theta_i + (1 - \mathbf{s}_i) \log(1 - \theta_i),$$

where $\mathbf{s}$ is a single state, $\theta_i$ is the sample mean calculated from $\mathcal{D}_t$ indicating the proportion of datapoints where location $i$ has been occupied by a block, and $s_i$ is a binary variable indicating the presence of a block at location $i$. If the blocks stack to the top, the game board resets, but the episode continues and the dataset $\mathcal{D}_t$ continues to accumulate states.

**SMiRL on *VizDoom* and *Humanoid*.**  In these environments the observations placed in the buffer are downsampled $10 \times 13$ single-frame observations for *VizDoom* environments and the full state for the *Humanoid* environments. We model $p(\mathbf{s})$ as an independent Gaussian distribution for each dimension in the observation. Then, the SMiRL reward can be computed as:

$$r_{\text{SMiRL}}(\mathbf{s}) = -\sum_i \left( \log \sigma_i + \frac{(\mathbf{s}_i - \mu_i)^2}{2\sigma_i^2} \right),$$

where $\mathbf{s}$ is a single state, $\mu_i$ and $\sigma_i$ are calculated as the sample mean and standard deviation from $\mathcal{D}_t$ and $\mathbf{s}_i$ is the $i^{th}$ observation feature of $\mathbf{s}$.

**SMiRL rewards**  We emphasize that the RL algorithm in SMiRL is provided with a standard stationary MDP (except in the VAE setting, more on that below), where the state is simply augmented with the parameters of the belief over states $\theta$ and the timestep $t$. We emphasize that this MDP is indeed Markovian, and therefore it is reasonable to expect any convergent RL algorithm to converge to a near-optimal solution. Consider the augmented state transition $p(s_{t+1}, \theta_{t+1}, t+1 | s_t, a_t, \theta_t, t)$. This transition model does not change over time because the updates to $\theta$ are deterministic when given $s_t$ and $t$. The reward function $R(s_t, \theta_t, t)$ is also stationary: it is in fact deterministic given $s_t$ and $\theta_t$. Because SMiRL uses RL in an MDP, we benefit from the same convergence properties as other RL methods.

However, the version of SMiRL that uses a representation learned from a VAE is not Markovian because the VAE parameters are not added to the state, and thus the reward function changes over time.. We find that this does not hurt results, and note that many intrinsic reward methods such as ICM and RND also lack stationary reward functions. This process is described in Algorithm 1.

**Entropic Environments**  We do not use entropic to mean that state transition probabilities change over time. Rather, it means that for any state in the environment, random disruptive perturbations may be applied to the state. In such settings, SMiRL seeks to visit state distributions $p(s)$ that are easy to preserve.

**VAE on-line training**  When using a VAE to model the surprise of new states, we evaluate the probability of the latent representations $\mathbf{z}$, as described in Section 2.3. The VAE is trained at the end of each episode on all data seen so far across all episodes. This means that the encoder $q_\omega(\mathbf{z}|bs)$ is changing over the course of the $SMiRL$ algorithm, which could lead to difficulty learning a good policy. In practice, the rich representations learned by the VAE help policy learning overall.

**Training parameters.**  For the discrete action environment (*Tetris* and *VizDoom*), the RL algorithm used is deep Q-learning (Mnih et al., 2013) with a target Q network. For the *Humanoid* domains, we use TRPO (Schulman et al., 2015). For *Tetris* and the *Humanoid* domains, the policies are parameterized by fully connected neural networks, while *VizDoom* uses a convolutional network. The encoders and decoders of the VAEs used for *VizDoom* and *Humanoid* experiments are implemented as fully connected networks over the same buffer observations as above. The coefficient for the KL-divergence term in the VAE loss was 0.1 and 1.0 for the *VizDoom* and *Humanoid* experiments, respectively.

Figure 6: Results for imitation in *Tetris*.

**Imitation Results**

