# OpenReview forum: "SMiRL: Surprise Minimizing RL in Entropic Environments"
_ICLR.cc/2020/Conference — Reject_

### Official Review · AnonReviewer2 · 2019-10-20
**Official Blind Review #2**

**Rating:** 3

**Review:**

The paper proposes a novel RL algorithm to minimize ‘surprise’, and the empirical results show the efficiency. The experiments are well-done jobs. However, there are several problems in other parts:
1. Some terms in Section 2 are too general, such as ‘surprise’ and ‘stable states’. Please define these more precise.
2. Please give an exact description of the MDP you consider about. It seems that the transition probabilities change by time as you mention ‘entropic environments’. So I guess it’s a finite horizon MDP?
3. What’s the final situations of the policy and density function? Please give some theoretical results such as convergence or asymptotic properties of the policy and the density functions. At least, the results can be derived under simple MDP and density function settings.
4. What’s the relationship between the density function you mention in your paper and state distributions in RL?
5. The agent trying to reach stable states sounds like exploration is discouraged. How do you explain? “Provably Efficient Maximum Entropy Exploration” by Elad et al. 2018 proposes to maximize negative log-likelihood of state distributions to encourage exploration, which is opposite to your setting. How do you think?


**Experience Assessment:**

I have published one or two papers in this area.

**Review Assessment: Checking Correctness Of Derivations And Theory:**

I assessed the sensibility of the derivations and theory.

**Review Assessment: Checking Correctness Of Experiments:**

I assessed the sensibility of the experiments.

**Review Assessment: Thoroughness In Paper Reading:**

I read the paper at least twice and used my best judgement in assessing the paper.

---

> ### Author Response · Authors · 2019-11-07
> **MDP definition and exploration.**
>
> The main concern raised in the review is about the definition of “entropic” and whether SMiRL learns in a well-defined MDP because of it. We do not use “entropic” to mean that state transition probabilities change over time. Rather, it means that for any state in the environment, random disruptive perturbations may be applied to the state. In such settings, SMiRL seeks to visit state distributions $p(s)$ that are easy to preserve. We will include more details in the paper to clarify this property. Additionally, we emphasize that the RL algorithm in SMiRL is provided with a standard stationary MDP (except in the VAE setting, more on that below), where the state is simply augmented with the parameters of the belief over states $\theta$ and the timestep $t$. We emphasize that this MDP is indeed Markovian, and therefore it is reasonable to expect any convergent RL algorithm to converge to a near-optimal solution. Consider the augmented state transition $p(s_{t+1}, \theta_{t+1}, t+1 | s_{t}, a_{t}, \theta_{t}, t )$. This transition model does not change over time because the updates to $\theta$ are deterministic when given $s_t$ and $t$. The reward function $R( s_{t}, \theta_{t}, t )$ is also stationary: it is deterministic given $ s_{t}$ and $\theta_{t}$. Because SMiRL uses RL in an MDP, we benefit from the same convergence properties as other RL methods. This description has been added to the paper to assist in making the method more clear to the reader.
>
> However, the version of SMiRL that uses a representation learned from a VAE is not Markovian because the VAE parameters are not added to the state, and thus the reward function changes over time. We find that this does not hurt results, and note that many intrinsic reward methods such as ICM and RND also lack stationary reward functions.
>
> Another concern in the review is whether SMiRL discourages exploration. SMiRL does not discourage exploration over its lifetime. When SMiRL is in a situation such that maintaining a stable $p_{\theta_{t}}(s)$ is difficult, then SMiRL will explore to learn the actions necessary to reach a more maintainable state distribution. In the paper, we outline this concept in Figure 1, where a robot must explore and learn how to build a house so that it can protect itself from changing weather in the future. The environments we use in the paper have entropy and noise; as a result, the agent must learn how to actively explore in order to reach and maintain homeostasis/equilibrium while battling entropy. We will include this more in-depth details in the paper.

---

> ### Author Response · Authors · 2019-11-13
> **Rebuttal Discussion**
>
> Dear Reviewer,
>
> Could you let us know if our response has addressed the concerns raised in your review? We would be happy to provide further revisions or experiments to address any remaining issues and would appreciate a response from you on the points that we raised.

---

### Official Review · AnonReviewer3 · 2019-10-21
**Official Blind Review #3**

**Rating:** 6

**Review:**

This paper proposes Surprise Minimizing RL (SMiRL), a conceptual framework for training a reinforcement learning agent to seek out states with high likelihood under a density model trained on visited states. They qualitatively and quantitatively explore various aspects of the behaviour of these agents and argue that they exhibit a variety of favourable properties. They also compare their surprise minimizing algorithm with a variety of novelty-seeking algorithms (which can be considered somewhat the opposite) and show that in certain cases surprise minimization can result in more desirable behaviour. Finally, they show that using surprise minimization as an auxiliary reward can speed learning in certain settings.

Minimizing surprise with RL presents challenges because, as the authors point out, exploring a surprising state in the present might minimize surprise in the future if the surprising state can be maintained (and therefore made unsurprising). To formulate this notion of surprise as a meaningful RL problem it is necessary to include some representation of the state likelihood in the agent's state representation; so that the agent can learn how it's actions affect not only the environment state but it's own potential for future surprise. This paper takes a pragmatic approach to this by training a density model on the full set of visited states which are reset at the start of each of a series of training episodes. The agent's policy is then conditioned on the parameters of the density model and the current size of the dataset.

I lean toward accepting this paper as an interesting initial step in applying the idea of surprise minimization to reinforcement learning. The proposed approach is simplistic in how they treat surprise, and therefore illustrative, but probably not practical. The experiments are also illustrative, and while it is clear that surprise minimization won't always generate useful behaviour, the paper doesn't overclaim in this regard. I found the motivation for being useful in natural environments, where maintaining some kind of homeostasis is often key, to be well presented. Given that a significant body of recent work focuses on novelty seeking as a means to guide exploration, I think it is a point worth making that there are many reasonable environments where the opposite behaviour is desirable.

I also found the paper to be quite well written and enjoyable to read overall.

General Comments:
Section 3, VizDoom: "The observation space for consists..."->"The observation space consists..."
Section 2.3: "...as a multivariate normal distribution..." it looks like covariance is not accounted for so wouldn't it more accurately be representing each component as an independent normal distribution?
Section 4.1: "...fireball explorations."->"...fireball explosions."?
Section 4.1: "...as shown in Figure Figure..."->"...as shown in Figure..."

Questions for the authors:
-Out of curiousity, what do the results look like when using SMiRL as a stability reward but without example trajectories?
-SMiRL is largely pitched as an alternative to novelty-seeking methods. But it seems to me novelty-seeking could be usefully combined since as you point out SMiRL may still have to explore to find stable states. Do you see such a combination as feasible or are the two methods fundamentally opposed?

**Experience Assessment:**

I have published one or two papers in this area.

**Review Assessment: Checking Correctness Of Derivations And Theory:**

N/A

**Review Assessment: Checking Correctness Of Experiments:**

I assessed the sensibility of the experiments.

**Review Assessment: Thoroughness In Paper Reading:**

I read the paper at least twice and used my best judgement in assessing the paper.

---

> ### Author Response · Authors · 2019-11-10
> **Additional experiments**
>
> We appreciate the comments on the work.
>
> Related to the reviewer's curiosity experiments, we have performed these and have added the results to the paper. The question of combining surprise and curiosity has been something we have considered. While on the surface, SMiRL minimizes surprise and curiosity approaches like ICM maximize surprise, they are in fact, not mutually incompatible. SMiRL surprise minimization is episodic, so in effect, combining it with novelty-seeking exploration uses exploration bonuses to help find better strategies for minimizing surprise. On the treadmill environment, we added a new experiment that shows that ICM and SMiRL rewards can be combined to achieve even better results (Figure 4(a) right “SMiRL + ICM” and on the updated webpage "Treadmill SMiRL + ICM"). The combination of methods leads to increased initial learning speed and producing a walking gait on that task.
>
> We have also run a version of the SMiRL as a stability reward experiment where $p(s)$ is not initialized with expert data (Figure 5(b,c) “Reward + SMiRL (no-expert)”). In this configuration, SMiRL improves on the number of falls during training and average task reward.
>
> We have also updated the paper to reflect the writing errors you found.

---

### Official Review · AnonReviewer1 · 2019-10-22
**Official Blind Review #1**

**Rating:** 6

**Review:**

Summary

This paper proposes a novel form of surprise-minimizing intrinsic reward signal that leads to interesting behavior in the absence of an external reward signal. The proposed approach encourages an agent to visit states with high probability / density under a parametric marginal state distribution that is learned as the agent interacts with its environment. The method (dubbed SMiRL) is evaluated in visual and proprioceptive high-dimensional "entropic" benchmarks (that progress without the agent doing anything in order to prevent trivial solutions such as standing and never moving), and compared against two surprise-maximizing intrinsic motivation methods (ICM and RND) as well as to a reward-maximizing oracle. The experiments demonstrate that SMiRL can lead to more sensible behavior compared to ICM and RND in the chosen environments, and eventually recover the performance of a purely reward-maximizing agent. Also, SMiRL can be used for imitation learning by pre-training the parametric state distribution with data from a teacher. Finally, SMiRL shows the potential of speeding up reinforcement learning by using intrinsic motivation as an additional reward signal added to the external task-defining reward.

Quality

As a practical paper, this work needs to be judged based on the quality of the experiments. I find the number of benchmarks and baselines sufficient. One major issue, that currently prevents me from voting for acceptance, is that experiments have not been conducted with enough seeds. I couldn't find any information in the paper regarding how many repetitions there are for each experiment. Also, most figures do not indicate any uncertainty measures (standard deviation, percentiles or the like)---some do, e.g. Figure (4), but it is not mentioned what type of uncertainty is depicted. One seed is certainly not enough to support the claims made by the authors---especially not the one that SMiRL can help improve RL in Figure (5). Figure (5a) clearly  does not draw a clear picture under one seed, and (5b) and (5c) require additional expert demonstrations. If experiments are repeated with more seeds and still support the claims, I am happy to increase my score to acceptance (presupposing that the discussion phase does not prevent acceptance for other reasons).

Clarity

The paper is clearly written and easy to follow. However, I do not like one aspect in the way the authors motivate their approach. The problem formulation starts with an MDP formulation. MDPs rely on a stationary reward signal and RL agents aim to optimize for future cumulative rewards based on these stationary reward signals. The authors propose to optimize for a non-stationary signal since the parametric state distribution changes over time. This itself is not uncommon and nothing controversial from a practical perspective. However, the statement that SMiRL agents seek to visit states that will change the parametric state distribution to obtain higher intrinsic reward in the future is controversial (see e.g. at the end of Section 2.1), because optimizing a non-stationary signal is outside the scope of the problem formulation. The reinforcement learning problem of maximizing intrinsic rewards does not know how the intrinsic reward signal is altered in the course of the future, i.e. how the parametric state distribution is updated. These statements should therefore be either adjusted accordingly, or the claims should be backed up theoretically rather than intuitively. On a minor note, I don't think that Figure (1) is necessary and the quote at the beginning of the paper might be better suited for a book chapter (but that is just my personal opinion).

Originality

I find the simple idea presented by the authors to minimize rather than maximize surprise quite original. However, I do not have much experience in the domain of intrinsic motivation and leave the judgement of originality to the other reviewers and the area chair. Also, some references might be missing regarding learning with intrinsic reward (e.g. empowerment).

Significance

The fact that a simple intrinsic reward, as presented by the authors, can lead to interesting behavior, as demonstrated by the experiments, is quite significant. Unfortunately, the experiments are not significant from a statistical perspective which is why I do not recommend acceptance at this stage (as mentioned above, depending on how the authors address this issue and the discussion period, I might change to acceptance).

Update

The authors have addressed my main concern regarding missing seeds. I therefore change to weak accept. But I am not happy how the authors responded to my concern regarding their motivation:
1.) The formulation that tries to justify their reasoning as given in the rebuttal was absent in the first version of this paper, but something along this line would have been necessary since the argument relies on a non-standard MDP formulation.
2.) I don't think the argumentation given in the rebuttal is correct. Imagine hypothetically the agent is in exactly the same augmented state s_t, a_t, \theta_t at two different points in time, e.g. in the first episode and after multiple episodes. In both cases, the collected states seen so far are going to be different, hence the optimization objective for learning a marginal state distribution is different, hence the parameter updates are different, hence the transitions are not stationary.
I do therefore encourage the authors to attenuate their wording.

**Experience Assessment:**

I have read many papers in this area.

**Review Assessment: Checking Correctness Of Derivations And Theory:**

I assessed the sensibility of the derivations and theory.

**Review Assessment: Checking Correctness Of Experiments:**

I assessed the sensibility of the experiments.

**Review Assessment: Thoroughness In Paper Reading:**

I read the paper at least twice and used my best judgement in assessing the paper.

---

> ### Author Response · Authors · 2019-11-13
> **More random seeds**
>
> We appreciate your time and comments on the work.
>
> The main concern in this review is the lack of multiple random seeds for the experiments shown in the paper. While it seems we missed describing this in the paper, all of the humanoid robot examples were already averaged over 3 seeds and the standard deviation is shown in Figures 4a, 4b, 5b and 5c. We have collected data for 5 seeds over all other environments in the paper (Figure3, 5a). We have included this extra data in the paper and a description of the seeds and standard deviations. The findings from this more in-depth analysis has not altered any claims in the paper.
>
> There was also concern raised by the review that the reward function is non-stationary. We emphasize that the RL algorithm in SMiRL is provided with a standard stationary MDP (except in the VAE setting), where the state is simply augmented with the parameters of the belief over states $\theta$ and the timestep $t$. We emphasize that this MDP is indeed Markovian, and therefore it is reasonable to expect any convergent RL algorithm to converge to a near-optimal solution. Consider the augmented state transition $p(s_{t+1}, \theta_{t+1}, t+1 | s_{t}, a_{t}, \theta_{t}, t )$. This transition model does not change over time because the updates to $\theta$ are deterministic when given $s_t$ and $t$. The reward function $R( s_{t}, \theta_{t}, t )$ is also stationary: it is deterministic given $ s_{t}$ and $\theta_{t}$. Because SMiRL uses RL in an MDP, we benefit from the same convergence properties as other RL methods. We have added details to make this more clear to the paper appendix.
>
> We have also added references to empowerment in section 5 and included more theoretical justification of our method and its properties to the appendix.

---

> ### Author Response · Authors · 2019-11-15
> **Response to  the update**
>
> Thank you for your update and helpful comments. We think there may be a misunderstanding of how we we calculate the augmented state.
>
> The augmented state includes the state $s_t$, the parameter $\theta_t$, and the timestep $t$.
> We define $\theta_t$ as the sufficient statistic over the history of states within an episode; $\theta_t$ does not incorporate information from previous episodes. Given two identical states transitions $(s_t, \theta_t, t),  a_t, (s_{t+1}, \theta_{t+1}, t+1)$ at the same timestep t in two different episodes, $\theta_{t+1}$ will be, by definition, updated in the same way. This is because $\theta_t$ and $t$ are the sufficient statistics for the distribution over states used for the reward.
>
> Because $\theta_t$ is independent across episodes and because the augmented state includes the timestep $t$, the transition dynamics T($ (s_{t+1}, \theta_{t+1}, t+1)| (s_t, \theta_t, t),  a_t$) are constant over the full training procedure, and the environment is a true MDP.

---

### Decision · Program_Chairs · 2019-12-19

**Decision:**

Reject

**Comment:**

This paper proposes augmentation of the state exploration strategy that is interesting and has a potential to lead to improvement. However, the current presentation makes it difficult to properly assess that. In particular, the way the authors convey both the underlying intuition and its implementation is fairly vague and does not build confidence in the grounding of the underlying methodology.